

# Feedforward pitch control for a 15-MW wind turbine using a spinner-mounted single-beam lidar

Wei Fu[1], Feng Guo[2], David Schlipf[2], and Alfredo Peña[1]

[1]Department of Wind and Energy Systems, Technical University of Denmark, Frederiksborgvej 399, 4000 Roskilde, Denmark
[2]Wind Energy Technology Institute, Flensburg University of Applied Sciences, Kanzleistraße 91-93, 24943 Flensburg, Germany

**Correspondence:** Wei Fu (weif@dtu.dk)

**Abstract.** Feedforward blade pitch control is one of the most promising lidar-assisted control strategies due to its significant improvement in rotor speed regulation and fatigue load reduction. A high-quality preview of the rotor-effective wind speed is a key element to control benefits. In this work, a single-beam continuous-wave or a pulsed lidar system is simulated in the spinner of a bottom-fixed IEA 15 MW wind turbine. The single-beam lidar can rotate with the wind turbine rotor and scan
the inflow with a circular pattern, which mimics a multiple-beam nacelle lidar at a lower cost. Also, the spinner-based lidar has an unimpeded view of the inflow without intermittent blockage from the rotating blade. The focus distance and the cone angle of the spinner-based single-beam lidar are optimized for the best wind preview quality based on a rotor-effective wind speed coherence model. Then, the control benefits of using the optimized spinner-based lidar are evaluated for an above-rated wind speed in OpenFAST with an embedded lidar simulator and virtual four-dimensional Mann turbulence fields considering
the wind evolution. Results are compared against those using a single-beam nacelle-based lidar. We found that the optimum scanning configurations of both CW and pulsed spinner-based single-beam lidars lead to a lidar scan radius of 0.6 of the rotor radius. Also, results show that a single-beam lidar mounted in the spinner brings much more control benefits (i.e., better rotor speed regulations and higher reductions of the damage equivalent loads on the tower base and blade roots) than the one based on the nacelle. The spinner-based single-beam lidar brings similar performance as a 4-beam nacelle lidar when used for
feedforward control.

## 1 Introduction

In the past decade, lidar-assisted wind turbine control (LAC) has received growing interest in the wind energy community. Among different control strategies, blade pitch feedforward control is one of the most promising LAC techniques, due to the significant improvement in the regulation of the rotor speed and the reduction of the fatigue loads compared to using conven-
tional feedback controllers alone (Canet et al., 2021). Whereas the feedback controller reacts to the wind disturbance after the effect of turbulent wind on the structure has occurred, the feedforward controller is able to utilize the preview information of the approaching wind provided by, e.g., lidars, which helps the turbine to react in advance. The collective pitch control strategy, in which the blades are controlled all together, uses the rotor-effective wind speed (REWS) as a key input to the feedforward controller.



In 2022, the installed prototype of the world's biggest wind turbine had a rated power of 15 MW. It has reached over 200 m in height and the rotor-swept area is equivalent to four soccer fields (Venditti, 2022). The inflow to wind turbines of such size cannot be measured by anemometers installed on a meteorological mast. The nacelle-mounted anemometers operate in the wake of the rotor and do not measure the free-stream wind speed. As remote sensing devices, forward-looking lidars mounted on the nacelle or the spinner of the wind turbines have a better sight of the wind approaching the rotor, and they can provide a high-quality wind preview. They are aligned with the wind turbine rotor and always track the incoming wind. Also, nacelle lidars can measure the inflow remotely at different locations over the rotor-swept area. The REWS estimated from a lidar system by combining the radial measurements over a full scan might more closely resemble the true REWS, which is the spatial average of the longitudinal wind velocities across the rotor disk (Schlipf et al., 2015a), than a point-wise anemometer. Therefore, they have the potential to deliver inflow characteristics that are better correlated with turbine signals (rotor speed, fatigue loads, etc.) than those derived from point-wise anemometers, e.g., cup and sonic anemometers.

Two types of nacelle lidar systems have been tested for wind turbine control, namely the continuous-wave (CW) and pulsed systems. The CW lidars usually measure at one focus distance at a time at a high sampling rate. Pulsed lidars are able to collect backscattered signals from several measurement ranges according to the response time, but they require typically long sampling periods. Both lidars have been reported useful for LAC (Mikkelsen et al., 2013; Kumar et al., 2015). Schlipf et al. (2014) found a decrease in the rotor speed variation during the above-rated operation of the CART2 using feedforward pitch control and a circularly-scanning pulsed lidar. Scholbrock et al. (2013) showed the mitigation on tower fore-aft loads using measurements from a three-beam pulsed lidar for the feedforward controller on the CART3. Scholbrock et al. (2015) achieved a reduction in yaw error using the circularly-scanning CW lidar replacing the turbine-based wind vane, etc. Although many other relevant studies are based on aero-elastic simulations (Harris et al., 2006a; Bossanyi et al., 2012; Simley et al., 2014), the results from the above experiments demonstrate improvements in wind turbine performance when using LAC (Simley et al., 2018).

The benefit of LAC needs to be balanced with the investment in using nacelle lidars. The simplest basic option is a single-beam staring lidar system. As the first field test of a nacelle-mounted lidar, Harris et al. (2006b) demonstrated that a single-beam CW lidar measuring at hub height is able to detect the fluctuations of the longitudinal velocity at 200 m upstream of a Nordex N90 wind turbine. Nevertheless, the measurement at a single location is not representative of the REWS.

Compared to the staring lidar mounted on the wind turbine nacelle, the single-beam lidar in the spinner can rotate with the rotor during turbine operation, scan a good portion of the inflow coming to the rotor disk, and reduce the cost of nacelle lidars relying on complex scanning patterns. Another advantage of using a spinner-based lidar, over a nacelle-mounted system, is the unimpeded view of the inflow without intermittent signal blockage by the blades, which increases data availability. A proof-of-concept field experiment was conducted by Mikkelsen et al. (2013), in which a ZephIR single-beam lidar system was deployed in the spinner of a NM80 2.3 MW wind turbine. They showed that the system is capable of measuring the upcoming wind and turbulence structure in real time. Based on a simulation study of the spinner-based CW lidar on the NREL 5 MW wind turbine, Simley et al. (2014) examined the accuracy of different measurement scenarios and found the best along-wind





component estimation at a lidar scan radius of 75% blade span, while the lidar provides the best blade-effective wind speed
estimation at 69% blade span.

This work aims at demonstrating the usefulness of a single-beam lidar for wind turbine feedforward control if the lidar is
mounted in the spinner compared to a nacelle-based system. Our reference wind turbine is the bottom-fixed variable-speed
collective-pitch-controlled IEA 15-MW turbine (design class 1B) with a rotor diameter of 240 m and a hub height of 150 m
(National Renewable Energy Laboratory, 2020). We consider both continuous-wave and pulsed Doppler lidars. Based on
the four-dimensional (4D) Mann turbulence model that considers wind evolution (Guo et al., 2022a), we optimize the focus
distance and the cone angle of the spinner-mounted single-beam lidar to achieve the highest coherence between the rotor- and
the lidar-estimated REWS. Then, through time-domain simulations using the 4D Mann turbulence fields with typical turbulence
parameters of near-neutral atmospheric stability conditions, the performance of the feedforward control using the optimized
lidar is evaluated. The ROSCO controller (Abbas et al., 2022) is used as the reference feedback controller. The simulations are
conducted in the open-source aero-elastic tool OpenFAST (National Renewable Energy Laboratory, 2022), and the results are
compared against those using a single-beam nacelle-based lidar.

This paper is organized as follows. Section 2 describes the background for this work including the turbulence spectral model,
the modeling of the wind evolution, the spinner-based lidar, and the wind preview quality. Section 3 introduces the set-up of
time-domain simulations. Section 4 shows the results of the lidar configuration optimization, which is followed by Section 5,
where we evaluate the performance of the feedforward control. Discussion of results is given in Section 6. Section 7 concludes
the work and provides the outlook.

## 2 Background

### 2.1 Mann turbulence spectral model

The three-dimensional wind field can be described by a vector field $\boldsymbol{u}(\boldsymbol{x}, t_0) = (u, v, w) = (u_1, u_2, u_3)$ at a given time $t_0$,
where $u, v, w$ are the horizontal along-wind, the horizontal lateral and the vertical wind components, respectively. The vector
$\boldsymbol{x} = (x, y, z)$ is the position vector defined in the right-handed Cartesian coordinate system. Using Reynolds decomposition, the
wind field can be decomposed into the mean wind speed $\boldsymbol{U} = \langle \boldsymbol{u}(x, 0, 0) \rangle = (U, 0, 0)$, where $\langle \cdot \rangle$ denotes ensemble averaging,
and the fluctuating components $(u', v', w')$. Assuming Taylor's frozen hypothesis (Taylor, 1938), the velocity fluctuations do
not change with time but propagate in the along-wind direction with a velocity equal to the mean wind speed. Therefore, the
wind field after a given time $\Delta t$ can be derived as

$$\boldsymbol{u}(x, y, z, t_0 + \Delta t) = \boldsymbol{u}(x - U\Delta t, y, z, t_0). \tag{1}$$

The wind field can also be expressed in the wavenumber domain using the three-dimensional Fourier transform:

$$\boldsymbol{u}(\boldsymbol{k}, t_0) = \frac{1}{(2\pi)^3} \int \boldsymbol{u}(\boldsymbol{x}, t_0) \exp(-\mathrm{i}\boldsymbol{k} \cdot \boldsymbol{x}) \mathrm{d}\boldsymbol{x}, \tag{2}$$





where $\boldsymbol{k} = (k_1, k_2, k_3)$ and $\int(\cdot)\mathrm{d}\boldsymbol{x} \equiv \int_{-\infty}^{\infty}\int_{-\infty}^{\infty}\int_{-\infty}^{\infty}(\cdot)\mathrm{d}x\mathrm{d}y\mathrm{d}z$. Denoting complex conjugate by * and the three velocity components by indices $i, j = 1, 2, 3$, the ensemble average of the Fourier coefficients is the spectral velocity tensor:

$$\langle u_i^*(\boldsymbol{k}, t_0)u_j(\boldsymbol{k}', t_0)\rangle = \Phi_{ij}(\boldsymbol{k})\delta(\boldsymbol{k} - \boldsymbol{k}'). \tag{3}$$

With the Dirac delta function $\delta(\cdot)$, Eq. (3) implies the homogeneity of the stochastic wind field, i.e., $\langle u_i^*(\boldsymbol{k})u_j(\boldsymbol{k}')\rangle$ is zero for $\boldsymbol{k} \neq \boldsymbol{k}'$. Here, we assume that the spectral tensor $\Phi_{ij}(\boldsymbol{k})$ can be described by the Mann model (Mann, 1994), in which, besides the wave number $k$, three adjustable parameters are used: $\alpha\varepsilon^{2/3}$, where $\alpha$ is the spectral Kolmogorov constant and $\varepsilon$ the turbulent energy dissipation rate, $L$, which is a length scale describing the size of the most energy-containing eddies, and $\Gamma$, which represents the turbulence anisotropy and distortion of the eddies from the vertical velocity shear in the atmospheric surface layer. The characteristics of the Mann model permit the modeling of three-dimensional spectra and coherence. The model is recommended by the IEC 61400-1 standard IEC (2019) for the calculation of wind turbine loads.

## 2.2 Temporal evolution of turbulence

Turbulence structures evolve when they approach the rotor. To consider the temporal evolution of turbulence, we assume that the stochastic field travels with the mean wind speed $U$ in the along-wind direction. However, we assume the turbulent eddies decay exponentially with time. The spectral velocity tensor $\Phi_{ij}$ then becomes space-time tensor $\Theta_{ij}$ (Guo et al., 2022a):

$$\Theta_{ij}(\boldsymbol{k}, \Delta t) = \exp\left(\frac{-\Delta t}{\tau_e(\boldsymbol{k})}\right)\Phi_{ij}(\boldsymbol{k}), \tag{4}$$

with

$$\langle u_i^*(\boldsymbol{k}, t_0)u_j(\boldsymbol{k}', t_0 + \Delta t)\rangle = \Theta_{ij}(\boldsymbol{k}, \Delta t)\delta(\boldsymbol{k} - \boldsymbol{k}'), \tag{5}$$

where $\tau_e$ is a new eddy lifetime that considers the temporal evolution. We also assume this eddy lifetime as in (Guo et al., 2022a):

$$\tau_e(\boldsymbol{k}) = \gamma\left[a(|\boldsymbol{k}|L)^{-1}\left((|\boldsymbol{k}|L)^{10} + 1\right)^{-\frac{2}{15}}\right], \tag{6}$$

where $\gamma$ is a coefficient that determines the strength of turbulence evolution. Guo et al. (2022a) and Guo et al. (2023) considered $\gamma \approx 400$ for near-neutral atmospheric stability conditions, and $\gamma \approx 200$ for stable atmospheric conditions.

The one-dimensional cross-spectra of all velocity fluctuations with separations $\Delta y$ and $\Delta z$ that consider evolution is then:

$$F_{ij}(k_1, \Delta t, \Delta y, \Delta z) = \int \Theta_{ij}(\boldsymbol{k}, \Delta t)\exp(\mathrm{i}(k_2\Delta y + k_3\Delta z))\mathrm{d}\boldsymbol{k}_\perp, \tag{7}$$

where $\int\mathrm{d}\boldsymbol{k}_\perp \equiv \int_{-\infty}^{\infty}\int_{-\infty}^{\infty}\mathrm{d}k_2\mathrm{d}k_3$. The one-point cross-spectra and auto-spectra of the velocity components can be obtained when the separations $\Delta y$ and $\Delta z$ are zero, and $i = j$ in Eq. (7). The magnitude squared coherence of all velocity components is

$$\mathrm{coh}_{ij}^2(k_1, \Delta t, \Delta y, \Delta z) = \frac{|F_{ij}(k_1, \Delta t, \Delta y, \Delta z)|^2}{F_{ii}(k_1, \Delta t = 0)F_{jj}(k_1, \Delta t = 0)}, \tag{8}$$





where

$$F_{ii}(k_1, \Delta t = 0) = \int \Phi_{ii}(\boldsymbol{k})\mathrm{d}\boldsymbol{k}_\perp. \tag{9}$$

### 2.3 Spinner-mounted single-beam lidar

In this work, we simulate a single-beam lidar system mounted in the wind turbine spinner. With an angle between the beam and the turbine's horizontal axis, the spinner-based lidar is able to scan the inflow in a circular pattern without signal blockage from the turbine blades or the nacelle, which is otherwise an issue in nacelle-mounted lidars. The beam orientation $\boldsymbol{n}$ can be expressed as

$$\boldsymbol{n}(\phi, \theta) = (n_1, n_2, n_3) = (-\cos\phi, \cos\theta\sin\phi, \sin\theta\sin\phi), \tag{10}$$

where $\phi$ is the half-cone opening angle, $\theta$ is the angle between the $y$-axis and the beam direction projected on the $y$-$z$ plane. The beam unit vector can also be expressed with the beam azimuth $\alpha$ and elevation angle $\beta$, which is used in the OpenFAST lidar simulator (Guo et al., 2022b):

$$\boldsymbol{n}(\alpha, \beta) = (-\cos\alpha\cos\beta, \sin\alpha\cos\beta, \sin\beta). \tag{11}$$

The four angles are marked in Figure 1. The rotor shaft of the reference wind turbine has a tilt angle of $6°$. Therefore, the lidar

beam unit vector is rotated around the $y$-axis. The red circles in Figure 1 indicate the scanning locations of the single-beam lidar before the rotation around the $y$-axis.

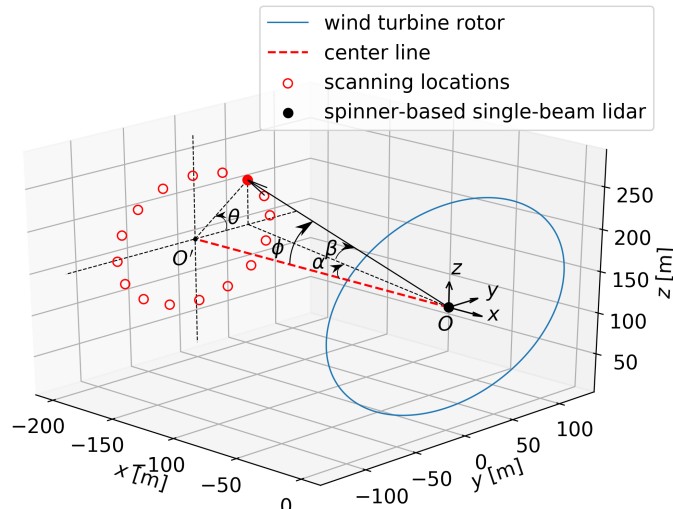

**Figure 1.** Scanning trajectory of the single-beam lidar in the IEA 15-MW wind turbine spinner. The used lidar angles are marked.

Since the typical feedforward collective pitch controller is only active at above-rated wind speeds, the rotational speed of the wind turbine has reached its rated value. For the reference wind turbine, the rated wind speed is $v_R = 10.59 \text{ ms}^{-1}$ and the





rated rotor speed is $\Omega_R = 7.56$ rpm. The turbine is controlled to maintain its rotor speed close to the rated value. Therefore,

the single-beam lidar needs $1/(7.56 \times 60) \approx 8$ s to complete a full scan. Assuming that the rotor speed is almost constant and the beam scanning locations are fixed with a sampling frequency $f_s = 4$ Hz, the spinner-based lidar can measure 32 radial velocities in one circular scan. Therefore, $\theta$ can be modelled as

$$\theta = \frac{2\pi}{60} \Omega_R i / f_s, \tag{12}$$

where $i = 1, 2, ..., 32$ is the beam index.

Assuming that the dominant radial velocity $v_r$ in the Doppler spectrum of radial velocities within the probe volume can be determined by the centroid method (Held and Mann, 2018; Fu et al., 2022), $v_r$ is the convolution of the lidar weighting function due to its probe volume $\varphi(s)$ and the wind components along the beam

$$v_r(\phi, \theta, f_d) = \int_{-\infty}^{\infty} \varphi(s) \boldsymbol{n}(\phi, \theta) \cdot \boldsymbol{u}[\boldsymbol{n}(\phi, \theta)(f_d + s)] \mathrm{d}s. \tag{13}$$

The weighting function of a CW lidar system is approximated by a Lorentzian function (Sonnenschein and Horrigan, 1971)

$$\varphi(s) = \frac{1}{\pi} \frac{z_R}{z_R^2 + s^2}, \tag{14}$$

where $s$ is the distance to the beam focus and $z_R$ is the Rayleigh length determined by the focus distance $f_d$, the laser wavelength $\lambda$, and the transmitted beam radius at the exit of the optical lens $r_b$

$$z_R = \frac{\lambda f_d^2}{\pi r_b^2}. \tag{15}$$

The Fourier transformation of Eq. (14) is

$$\hat{\varphi}(\boldsymbol{k}, \boldsymbol{n}) = \exp(-|\boldsymbol{k} \cdot \boldsymbol{n}| z_R). \tag{16}$$

For pulsed systems, we assume the weighting function has a Gaussian-shape parameterized by a standard deviation $\sigma_L$ (Cariou, 2013)

$$\varphi(s) = \frac{1}{\sigma_L \sqrt{2\pi}} \exp\left(-\frac{s^2}{2\sigma_L^2}\right), \tag{17}$$

with

$$\sigma_L = \frac{W_L}{2\sqrt{2\ln 2}}, \tag{18}$$

where $W_L$ is the Full-Width at Half Maximum (FWHM). The Fourier transform of Eq. (17) is

$$\hat{\varphi}(\boldsymbol{k}, \boldsymbol{n}) = |\boldsymbol{k} \cdot \boldsymbol{n}| \exp\left(-|\boldsymbol{k} \cdot \boldsymbol{n}|^2 \frac{\sigma_L^2}{2}\right). \tag{19}$$

It can be seen from Eq. (15) and Eq. (18) that the probe volume of CW lidars increases with the square of the focus distance, whereas it is constant at any range for pulsed systems. In our study, we assume $\lambda = 1.565 \ \mu$ m, $r_b = 28$ mm and $W_L = 30$ m

(Peña et al., 2016).





The weighting functions need to be truncated and discretized to simulate lidar measurements in turbulence boxes of finite length. We discretize Eq. (14) with a resolution of $\Delta s = 0.1z_\mathrm{R}$ considering $s_\mathrm{max} = 6z_\mathrm{R}$ and $s_\mathrm{min} = -6z_\mathrm{R}$. Similarly, for Eq. (17), we use $s_\mathrm{max} = 1.5W_\mathrm{L}$ and $s_\mathrm{min} = -1.5W_\mathrm{L}$ with a resolution of $\Delta s = 2.5$ m (around $0.08W_\mathrm{L}$). The discretized weights are normalized to have the sum equal to one. Since $W_\mathrm{L} = 2z_\mathrm{R}$, the pulsed lidar probe volume is more compact and centralized than

that of the CW. Figure 2 compares the truncated theoretical weighting functions of the two lidar systems measuring at different ranges. To illustrate the two types of weighting functions, the weights in Figure 2 are normalized by the maximum values. In our case, the pulsed lidar has a similar FWHM with the CW lidar focusing at $155$ m.

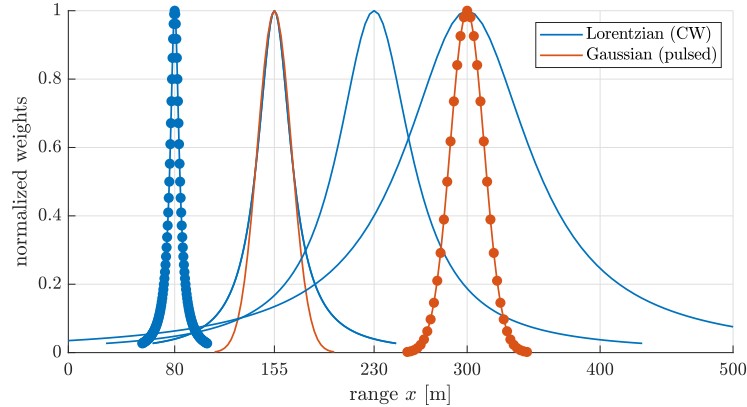

**Figure 2.** Weighting functions of the CW lidar measuring at 80 m, 155 m, 230 m, 300 m and the one of the pulsed lidar measuring at 155 m, 300 m. The weights are normalized by the maximum values for illustration purposes. The blue and red markers indicate the discretization of the functions.

The amount of truncation needs to be balanced between a realistic probe volume and the limited size of the virtual wind fields. The truncation and discretization influence the amount of turbulence attenuation by the probe volume. However, the

small turbulent eddies do not greatly impact the coherence of the REWS, since the spatial averaging by the rotor disk has a similar filtering effect on the true REWS.

### 2.4 Lidar wind preview quality

#### 2.4.1 Rotor-effective wind speed from the wind turbine

If the yaw misalignment is neglected, the true REWS is the spatial average of the longitudinal velocities $u$ across the rotor-swept

area defined by the rotor radius $R$ (Schlipf et al., 2015a):

$$u_\mathrm{RR}(x) = \frac{1}{\pi R^2} \iint\limits_{\text{rotor}} u(\boldsymbol{x}) \mathrm{d}y \mathrm{d}z. \tag{20}$$

Held and Mann (2019) demonstrated that this REWS can be rewritten as

$$u_\mathrm{RR}(x) = \int u(\boldsymbol{k}) e^{ik_1 x_1} \frac{2J_1(\kappa R)}{\kappa R} \mathrm{d}\boldsymbol{k}, \tag{21}$$



where $\kappa = \sqrt{k_2^2 + k_3^2}$ and $J_1$ is the Bessel function of the first kind. Held and Mann (2019) also showed that the auto-spectrum
of $u_{\mathrm{RR}}$ is

$$S_{\mathrm{RR}}(k_1) = \int_{-\infty}^{\infty} \Phi_{11}(\boldsymbol{k}) \frac{4J_1^2(\kappa R)}{\kappa^2 R^2} \mathrm{d}\boldsymbol{k}_\perp. \tag{22}$$

### 2.4.2 Rotor-effective wind speed estimated by the lidar

Assuming that the turbine yaw misalignment is negligible, the center line of the lidar scanning trajectory is on the turbine
rotation axis, and $v$ and $w$ are considered to be zero, the $u$ component can be estimated directly from the lidar measurements.
The lidar-estimated REWS is the mean of the along-wind component retrieved from the radial velocities along the beam:

$$u_{\mathrm{LL}}(t) = \sum_{i=1}^{N_b} \frac{1}{N_b n_{i1}} v_{\mathrm{r},i}(t), \tag{23}$$

where $N_{\mathrm{b}}$ is the number of measurements over a full scan and $n_{i1}$ is the first element in the unit vector of the $i^{\mathrm{th}}$ measurement.

Because the longitudinal wind evolution is the most important factor for control, and the considered lidars in this work only
measure at a single plane, the wind evolution between each measurement in a full scan is not considered, which should have
only a marginal effect on our optimization. The auto-spectrum of the lidar-estimated REWS is (Guo et al., 2022a)

$$S_{\mathrm{LL}}(k_1) = \sum_{i,j=1}^{N_b} \sum_{l,m=1}^{3} \frac{1}{N_b^2 n_{i1} n_{j1}} \int n_{il} n_{jm} \Phi_{lm} \exp(\mathrm{i}\boldsymbol{k} \cdot (\boldsymbol{x}_i - \boldsymbol{x}_j)) \hat{\varphi}(\boldsymbol{k} \cdot \boldsymbol{n}_i) \hat{\varphi}(\boldsymbol{k} \cdot \boldsymbol{n}_j) \mathrm{d}\boldsymbol{k}_\perp, \tag{24}$$

where $\boldsymbol{x}_i$ denotes the position vector of the lidar measurement, $n_{il}$ stands for the $l^{\mathrm{th}}$ element in the unit vector $\boldsymbol{n}$ of the $i^{\mathrm{th}}$
measurement.

For control purposes, the lidar scanning strategy is considered optimal, if it provides REWS estimates that correlate the best
with the true REWS sensed by the rotor disk. Considering the turbulence evolution from lidar measurement planes to the rotor
plane, the cross-spectrum between $u_{\mathrm{RR}}$ and $u_{\mathrm{LL}}$ can be expressed as (Guo et al., 2022a)

$$S_{\mathrm{RL}}(k_1) = \sum_{i=1}^{N_b} \sum_{l=1}^{3} \frac{1}{N_b n_{i1}} \int n_{il} \Theta_{l1}(\boldsymbol{k}, \Delta t_i) \hat{\varphi}(\boldsymbol{k} \cdot \boldsymbol{n}_i) \exp(\mathrm{i}(k_2 x_{i2} + k_3 x_{i3})) \frac{2J_1(\kappa R)}{\kappa R} \mathrm{d}\boldsymbol{k}_\perp, \tag{25}$$

where $\Delta t_i$ denotes the time needed for the turbulence field to travel from a lidar plane to the rotor plane, given a good estimation
by their longitudinal separation divided by the mean along-wind speed, i.e., $\Delta t_i = |\Delta x_{iR}|/U$.

### 2.4.3 Rotor-effective wind speed coherence

The wind preview quality can be evaluated by the magnitude squared lidar-rotor REWS coherence (Schlipf, 2016; Simley et al.,
2018)

$$\gamma_{\mathrm{RL}}^2(k_1) = \frac{|S_{\mathrm{RL}}(k_1)|^2}{S_{\mathrm{RR}}(k_1) S_{\mathrm{LL}}(k_1)}, \tag{26}$$





which has a value between 0 and 1. The measurement coherence bandwidth (MCB) is defined as the wave number $k_{0.5}$ where
$\gamma_{\mathrm{RL}}^2$ drops below 0.5. The corresponding frequency can be calculated by $f_{0.5} = k_{0.5}U/(2\pi)$. The larger the MCB, the better the wind preview quality. Therefore, maximizing the MCB is the goal of lidar trajectory optimization.

To evaluate the lidar wind preview quality, the so-called 'smallest detectable eddy size' $d_{\mathrm{eddy,min}}$ is used by control engineers, which is the size of the eddies that can still be detectable by the lidar with the $50\%$ coherence assuming turbulence isotropy (Schlipf et al., 2018)

$$d_{\mathrm{eddy,min}} = \frac{2\pi}{k_{0.5}}. \tag{27}$$

The smallest detectable eddy size is inversely proportional to the MCB. To have a measure that is independent of the rotor size, the $d_{\mathrm{eddy,min}}$ can be normalized by the rotor diameter of the reference wind turbine. A normalized $d_{\mathrm{eddy,min}}$ close to $1D$ indicates a very good lidar configuration for the purpose of fatigue load reduction, while a value between $1.5D$ and $2D$ is satisfying.

The wind preview quality of the considered lidar configurations is directly calculated for the reference wind turbine in
the frequency domain using Eqs. (22), (24) and (25) instead of using time-domain simulations, which greatly reduces the computational effort and provides a more accurate MCB value compared to that estimated from simulated spectra of coherence in the time domain. Then, the controller performance using the optimal lidar configurations is evaluated using time domain aero-elastic simulations with Mann turbulent wind fields.

## 3    Time-domain simulation set-up

### 3.1    Simulation environment

The time-domain aero-elastic simulations are performed for the IEA 15-MW wind turbine using the open-source tool Open-FAST (National Renewable Energy Laboratory, 2022), in which a lidar simulator is embedded. Using the latest version of the OpenFAST lidar simulator (see Guo et al., 2022b, for more details), the probe volume, the turbine nacelle motion, and the turbulence evolution are included. The weighting function of the probe volume is given in discrete points as explained in
Section 2.3.

The four-dimensional stochastic turbulence fields are generated by the 4D Mann turbulence generator developed by Guo et al. (2022a). The turbulence fields have model parameters $\alpha\varepsilon^{2/3} = 0.2882$ $\mathrm{m}^{4/3}$ $\mathrm{s}^{-2}$, $L = 49$ m and $\Gamma = 3.1$, which are typical of near-neutral atmospheric conditions and corresponding to the IEC class 1B with a turbulence intensity of $\approx 15\%$ at the mean wind speed of $18$ $\mathrm{ms}^{-1}$. The mean wind field $\boldsymbol{U} = (U_{\mathrm{ref}}, 0, 0)$ at the turbine hub height and a power law shear profile
with a shear exponent of $0.14$, i.e., $U(z) = U_{\mathrm{ref}}\left(\frac{z}{z_{\mathrm{HH}}}\right)^{0.14}$, is added upon the turbulence boxes, where $z_{\mathrm{HH}}$ is the turbine hub height. The turbulence box has dimensions of $4096 \times 64 \times 64$ grid points in the $x$, $y$, and $z$ directions, respectively. The grid size in $y$ and $z$ directions are both $4.5$ m to cover the whole rotor disk and the tower in the vertical direction, while the resolution in the $x$ direction is $\Delta x = 0.5U_{\mathrm{ref}}$. All simulations are performed for a single wind speed of $U_{\mathrm{ref}} = 18$ $\mathrm{ms}^{-1}$. The blade, tower and generator degree of freedoms (DOFs) are enabled.



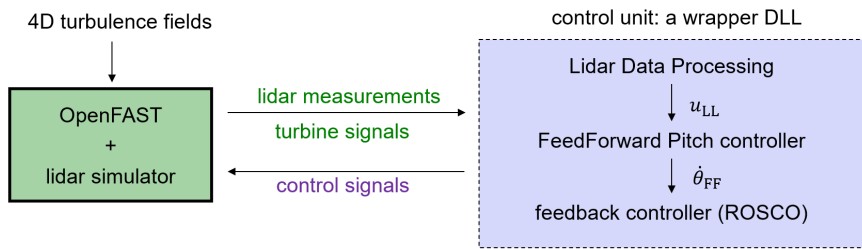

**Figure 3.** Structure of the communication interface between the OpenFAST and the controller dynamic-link library chain.

As illustrated in Figure 3, the turbulent wind disturbs the turbine. The turbine-lidar unit delivers lidar radial velocities and simultaneous turbine signals (generator speed and pitch angle) to the control unit, which then sends control signals (generator torque and demanded pitch angle) back to the turbine to demand control actions. Therefore, without the feedforward controller that relies on the wind preview, the feedback controller calculates control demands based on the past turbine signals and reacts to the disturbance only after the aerodynamic impact on the turbine structure has occurred. The feedforward controller utilizes

the lidar-estimated preview information and assists the feedback controller to react in advance. Since OpenFAST can only refer to a single dynamic-link library (DLL) as the control unit, a wrapper DLL is configured to encapsulate and call the lidar data processing, feedforward pitch controller, and feedback controller (ROSCO (Abbas et al., 2022)) sequentially in order to exchange signals with OpenFAST (Guo et al., 2023). The three subunits are introduced in the following subsections.

### 3.2   Lidar data processing

The simulated spinner-based lidar completes a full scan in approximately $8$ s with a sampling frequency of $4$ Hz. Therefore, the latest 32 measurements are collected to reconstruct the REWS using Eq. (23), and the reconstructed REWS is updated every $0.25$ s. In frequency-domain optimization, the beam scanning locations in the circular pattern are assumed to be fixed, while in time-domain simulation, the beam scanning locations depend on the rotor azimuth positions and nacelle motions in real time.

In practice, the REWS estimated from the lidar measurements is not perfectly correlated with the real one sensed by the

rotor. Therefore, a filter needs to be applied to the lidar-estimated REWS before using it for the feedforward controller to avoid unnecessary and harmful reactions from the pitch actuator. Here, a first-order Butterworth low-pass filter is applied

$$G_{\text{filter}}(s) = \frac{\omega_{\text{cutoff}}}{s + \omega_{\text{cutoff}}}, \tag{28}$$

with a cutoff angular frequency $\omega_{\text{cutoff}} = 2\pi f_{\text{cutoff}} = k_{\text{cutoff}} U_{\text{ref}}$, which is calculated from the cut-off wavenumber $k_{\text{cutoff}}$ where the theoretical REWS measurement transfer function drops at $-3$ dB (Schlipf, 2016; Guo et al., 2023) and $s$ is the complex

frequency. The theoretical REWS transfer function is calculated from Eqs. (25) and (24)

$$G_{\text{RL}} = \frac{|S_{\text{RL}}(f)|}{S_{\text{LL}}(f)}. \tag{29}$$





The low pass filtering usually delays a signal due to the frequency-depending phase shift. For the first-order filter, the time delay $T_{\text{filter}}$ is approximated by

$$T_{\text{filter}} = \frac{\arctan\left(\frac{f_{\text{delay}}}{f_{\text{cutoff}}}\right)}{2\pi f_{\text{delay}}}, \tag{30}$$

where $f_{\text{delay}}$ is the interested frequency (in our case 0.025 Hz), in which the simulated rotor speed spectrum by the feedback-only control has its highest energy. Therefore, the higher the cutoff frequency, the more useful information is available in the lidar-estimated REWS signals, and less time is needed for filtering the signal.

### 3.3 Feedforward controller

The feedforward controller is designed to stabilize the rotational speed in the changing inflow wind speed by demanding an
additional pitch angle $\theta_{\text{FF}}$ before the disturbance hits the rotor. In this way, the rotor speed acceleration $\dot{\Omega}$ caused by the wind speed fluctuations can be compensated by the additional pitch angle.

The design of the feedforward controller follows the methodology given in Schlipf (2016) and Guo et al. (2023). Considering a reduced-order model of the wind turbine with a single rotor rotation DOF:

$$J\dot{\Omega} = M_a(u_{\text{RR}}, \Omega, \theta_{\text{p}}) - M_{\text{G}}, \tag{31}$$

with

$$M_a = \frac{1}{2}\rho\pi R^2 \frac{c_{\text{P}(\lambda,\theta_{\text{p}})}}{\Omega} u_{\text{RR}}^3 \text{ and } \lambda = \frac{\Omega R}{u_{\text{RR}}}, \tag{32}$$

where $J$ is the rotor inertia, $\theta_{\text{p}}$ is the blade pitch angle, $c_{\text{P}}$ is the turbine power coefficient, $\lambda$ is the tip speed ratio, $M_a$ is the aerodynamic torque and $M_{\text{G}}$ is the generator torque. The aerodynamic effect on the rotational speed change can be canceled out if $M_a(u_{\text{RR}}, \Omega, \theta_{\text{p}}) = M_{\text{G}}$. Therefore, by changing the pitch angle, the aerodynamic torque is adjusted to be close to the rated
value of the generator torque. The feedforward pitch angle $\theta_{\text{FF}}$ should follow the static pitch curve $\theta_{\text{FF}} = \theta_{\text{p,ss}}(u)$, which can be obtained by steady-state simulations with a feedback controller and the uniform and constant wind of all speeds between cut-in and cut-off, as shown in Figure 4. At the cut-in wind speed, the blades have an initial pitch angle. The pitch angle first decreases to make the best use of the incoming wind, and increases after reaching the rated wind speed of $10.59 \text{ ms}^{-1}$. Thus, a feedforward pitch rate $\dot{\theta}_{\text{FF}}$ can be calculated using the derivation of the static pitch curve (see Schlipf, 2016, Chapter 6.1.1 for
more details). The feedforward pitch rate is used in the integrator block of the feedback controller.

### 3.4 Feedback controller

The modular Reference Open-Source COntroller (ROSCO) developed by Abbas et al. (2022) for fixed and floating wind turbines is used as the feedback controller in this work. The feedback controller contains two parts: a torque controller, which mainly regulates the generator torque $M_{\text{G}}$ to maximize the energy yield in below-rated wind speeds and keeps the power steady
in above-rated wind speeds, and a collective blade pitch controller, which maintains the rated generator speed in the fluctuating wind by changing the blade pitch angle.





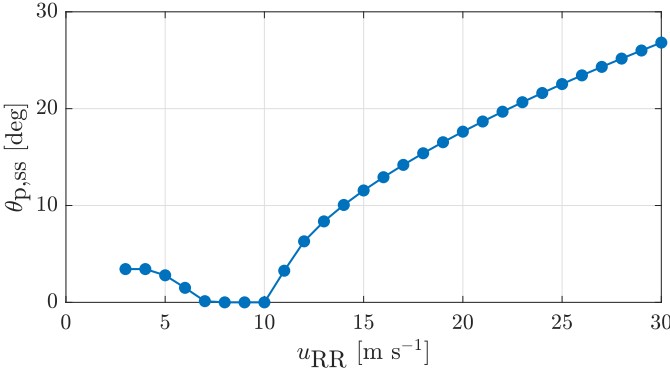

**Figure 4.** Static pitch curve of the bottom-fixed IEA 15-MW wind turbine performed with ROSCO in OpenFAST.

The baseline collective blade pitch controller is achieved by a proportional-integral controller described in Jonkman et al. (2009). Therefore, the calculated pitch angle is

$$\theta_{\text{FB}} = k_{\text{p}} \Delta\Omega + K_{\text{I}} \int_0^t \Delta\Omega \, \mathrm{d}t, \tag{33}$$

where $k_{\text{p}}$ is the proportional gain, $K_{\text{I}}$ the integral gain, $\Delta\Omega = \Omega_{\text{G,rated}} - \Omega_{\text{G}}$ the difference between the contemporary generator speed and its rated value, and $s$ the complex frequency. The default values of the feedback controller gains are used in this study.

The integral block of the feedback controller uses the feedforward pitch rate $\dot{\theta}$ passed by the feedforward controller. This gives the total demanded pitch angle $\theta_c$ as

$$\theta_c = \theta_{\text{FB}} + \int_0^t \dot{\theta}_{\text{FF}} \, \mathrm{d}t. \tag{34}$$

Then, the pitch actuator moves the blades according to the demanded pitch angle. The pitch actuator is modelled as a second-order damper system with the cut-off frequency of $1.5708 \,\text{rad}\,\text{s}^{-1}$ and the damping ratio of $0.707$ (Abbas et al., 2022), so the pitch actuation takes $T_{\text{pitch}} \approx 0.9 \,\text{s}$ for frequencies lower than $0.04$ Hz for the reference wind turbine.

### 3.5 Buffer time of REWS signal

To synchronize the pitch actuation with the REWS interacting with the turbine, the preview signal is usually buffered with a suitable time $T_{\text{buffer}}$. $T_{\text{buffer}}$ is calculated from the advection time of the wind field from the lidar measurement plane to the rotor plane $T_{\text{lead}} = \Delta x / U_{\text{ref}}$, the averaging time of the lidar raw measurement (half of a full scan time $T_{\text{scan}}$), the time consumed by the low-pass filter $T_{\text{filter}}$, and the pitch actuator delay $T_{\text{pitch}}$ (Schlipf, 2016):

$$T_{\text{buffer}} = T_{\text{lead}} - \frac{1}{2} T_{\text{scan}} - T_{\text{filter}} - T_{\text{pitch}}. \tag{35}$$



To ensure the feedback-feedforward combined controller has enough time to react to the wind disturbance before the wind hits
the rotor, $T_{\text{buffer}}$ has to be larger than zero. Since $T_{\text{lead}}$ and $T_{\text{filter}}$ are influenced by the lidar scanning trajectory, $T_{\text{buffer}} > 0$ s is a
constraint to select the optimal configuration.

## 4 Optimization of lidar configuration for wind preview quality

We optimize the scanning locations of the spinner-based single-beam lidar to achieve the best wind preview quality. Based
on the coherence model given in Section 2.4, we calculate the analytical MCB (also written as $k_{0.5}$) in the frequency domain
for different combinations of the lidar measurement range along the $x$-direction and the half-cone opening angle $\phi$. The focus
distance can be calculated from the measurement range by $f_{\text{d}} = x/\cos\phi$. For simplicity, only a single measurement range is
considered for both CW and pulsed lidars in this work, although measurements from multiple ranges can be obtained simulta-
neously using pulsed lidars. The lidar configuration is considered to be optimal when the highest MCB is achieved. Meanwhile,
the selected lidar configuration needs to give a positive buffer time, as described in Section 3.5, so that the controllers have
enough time to react to the changing wind.

The optimization is done assuming a mean wind speed of $18$ ms$^{-1}$. The optimization results are shown in Figure 5(a)
and (c) for CW and pulsed spinner-based lidars, respectively. Although measuring at $160$ m and $\phi = 24°$ with a CW lidar
provides the highest MCB in our optimization, it is not usable due to a negative buffer time. Therefore, the optimum scanning
configuration of the single-beam lidar in a CW system is selected at $x = 190$ m and $\phi = 21°$, and the one for the pulsed system
is at $x = 220$ m and $\phi = 18°$. The results indicate that the CW lidar gives a better wind preview when it measures closer to
the rotor with a wider angle compared to the pulsed lidar. This is expected since the probe volume filtering effect becomes
more influential for CW lidars the further the measurement range. The best range-opening angle combinations of both CW and
pulsed lidars result in a scan radius at approximately $72$ m ($0.6R$) (see Figure 5(b) and (d)). Due to the rotor shaft tilt angle, the
lidar scanning area is at the middle-top part of the rotor plane. With the optimum configuration, both types of spinner-mounted
single-beam lidars can achieve a maximum $k_{0.5}$ of more than $0.014$ m$^{-1}$ corresponding to a $d_{\text{eddy,min}}$ smaller than $1.87D$, while
the nacelle-based single-beam lidar achieves only approximately $0.005$ m$^{-1}$ (the single-point measurements provide $k_{0.5}$ that
are almost constant but slightly reduce with further measurement ranges).

With the same turbulence characteristics, the mean wind speed does not have a large impact on the modeling of REWS
coherence but is important for the selection of the optimum lidar configurations due to its impact on the buffer time. In our
case, the selected configuration of CW lidar gives a very short buffer time ($0.7$ s) indicating the measurement distance will be
too close for controllers to react if the wind speed is higher than $19$ ms$^{-1}$. Measuring at $220$ m and $\phi = 18°$ with a pulsed
lidar gives a buffer time of $1.62$ s, and the controller would have enough time to react for a mean wind speed below $20$ ms$^{-1}$.
A larger measurement range should be selected for both types of lidars if the full wind speed range (up to the wind turbine
cut-off wind speed of $25$ ms$^{-1}$) is considered. When the measurement range increases from the optimum point, the MCB
could decrease. Consequently, the low-pass filter will have a lower cut-off frequency and will need a longer time to process the

**Figure 5.** Left: optimization of the range $x$ and half-cone opening angle $\phi$ of the spinner-based lidar based on coherence model. The selected optimum configurations at a mean wind speed of $18\ \mathrm{ms^{-1}}$ are marked in a red circle. Right: The scanning pattern of the selected optimum configurations.





lidar measurement. Therefore, it is essential to estimate the REWS coherence for the selected scanning pattern and design the control unit accordingly.

Time domain simulations were executed in OpenFAST with the embedded lidar simulator, the optimal configurations of both
lidars given in Figure 5(b)(d), the feedback-feedforward controller and 4D Mann turbulence fields with a mean wind speed of $18\,\mathrm{ms}^{-1}$. To ensure statistical convergence of 10-min simulations, 21 realizations (seeds) of the same turbulence fields are used (Liew and Larsen, 2022). The time series of the filtered REWS $u_{\mathrm{LL}}$ is collected from the outputs of the feedforward controller, and the real REWS $u_{\mathrm{RR}}$ is calculated from the virtual turbulence fields by averaging the along-wind time series among the rotor swept area. Simulations with similar set-ups are performed using the nacelle-based lidar. The nacelle-based CW lidar is
simulated considering a measurement range of $200\,\mathrm{m}$ so that the controller has enough time to react to the turbulent wind with a mean wind speed of $18\,\mathrm{ms}^{-1}$ (it takes longer to filter the REWS signal estimated from the nacelle-based than the spinner-based lidar due to the low MCB). Figure 6 compares the REWS coherence and transfer functions from time-domain simulations and those calculated in the frequency domain using the method presented in Section 2.4. Results of the CW and pulsed types of lidar are shown in the upper and lower panels, respectively.

Comparing the left plots (spinner-based) with the right plots (nacelle-based) in Figure 6, we see that the coherence in terms of the $k_{0.5}$ has been improved a lot by using the optimized lidar in the spinner. Overall, the simulated REWS coherence fits with the analytical models, which indicates that the scanning configurations optimized in the frequency domain are also providing the best wind preview in the time domain. Some noise appears at high frequencies due to the spectra estimation process.

## 5 Feedforward control benefits

The benefits of using the feedforward pitch controller are evaluated in this section. Time-domain simulations are performed using the optimized lidar in the spinner and on the nacelle, respectively, first with the feedback controller only, and then with the feedback-feedforward combined controller. Simulations in each scenario are executed using turbulence fields with the same turbulence characteristics for 21 different seeds (Liew and Larsen, 2022). Therefore, for lidar in CW and pulsed systems, respectively, $2 \times 2 \times 21$ simulations are carried out. All DOFs of the 15-MW reference wind turbine are enabled and
no wave impacts are simulated. The simulation time is $640\,\mathrm{s}$ in total, in which the first $40\,\mathrm{s}$ is the transient and excluded from the analysis. Then, the spectra of the rotor speed, the tower base bending moment and the blade root bending moment are calculated from the simulated time series. Here, only results of CW lidars are shown, since similar results are found for pulsed lidars.

The analytical spectrum of the rotor speed using feedforward control is modelled as

$$S_{\Omega\Omega} = |G_{\Omega u_{\mathrm{LL}}}|^2 S_{\mathrm{RR}}(1 - \gamma_{\mathrm{RL}}^2), \tag{36}$$

where $G_{\Omega u_{\mathrm{LL}}}$ is the closed-loop transfer function from the REWS to the rotor speed, which consists of the linearized wind turbine model, the pitch actuator, the control units and the generator torque controller (Schlipf et al., 2015b).

Results are shown in Figure 7, in which the left panels are from spinner-based lidar and the right panels are from the nacelle-based lidar. The benefits of using feedback-feedforward control (FBFF) compared to the feedback-only (FB-only) case are







**Figure 6.** Coherence of the REWS using the optimal single-beam lidar (a)(c) in the spinner and (b)(d) on the nacelle. Upper panel for CW lidars, lower panel for pulsed lidars. Simulation results are averaged from 21 wind field realizations.





well visible mainly at low frequencies. This is expected since the low-frequency range is where the lidar wind preview signal correlates well with the real REWS. In Figure 7(a) and (b), the simulated rotor speed spectra fit well with the analytical one for the frequency range below $0.2$ Hz (below the 1P of the turbine). Significant reductions of the rotor speed variations are achieved using the spinner-based configuration compared to the nacelle-based one. Furthermore, within the low-frequency range, higher load reductions on the tower-base fore-aft (below $0.07$ Hz) and blade-root flap-wise directions (below $0.1$ Hz)

can be seen using the spinner-based lidar.

  The standard deviation of the rotor speed and the fatigue loads, i.e., damage equivalent loads (DELs) of the tower-base and blade-root bending moments are calculated from the time series. To estimate the DELs, the rain flow counting method introduced by Matsuichi and Endo (1968) is applied. The DELs are based on a reference number of cycles of $2 \times 10^6$ and a turbine lifetime of 20 years. Wöhler exponents of 4 and 10 are used for the tower-base fore-aft and blade-root flap-wise

bending moments respectively, as described in (Schlipf, 2016). Statistically, by using FBFF with the single-beam CW lidar in the spinner instead of on the nacelle, the reduction of the mean rotor speed standard deviation is improved from 13.8% to 47.4%, and the reduction of the tower-base fore-aft bending moment DEL increases from 1.0% to 4.3%. The strategy also brings 3.1% reduction to the blade-root flap-wise moment DEL. Since the default feedback controller parameters are adopted, the DEL reductions can be further improved by optimizing the controller gains (Schlipf et al., 2018).

Similar results and trends are seen from the simulations using the pulsed lidar, which are summarized in Table 1. We have also optimized the scanning pattern of a 4-beam CW nacelle lidar, which provides a MCB around $0.011$ m$^{-1}$ measuring at 220 m with $\phi = 15°$. The optimized 4-beam nacelle lidar is applied and simulated with 21 realizations of the same turbulence fields. Results in Table 1 show that the control benefits gained using the spinner-based single-beam lidar are larger than those we can achieve using the same lidar on the nacelle and that the benefits using a spinner-beam single-lidar are of a similar level

to those using a 4-beam system.

| reductions | spinner (CW) | nacelle (CW) | spinner (pulsed) | nacelle (pulsed) | 4-beam nacelle (CW) |
|---|---|---|---|---|---|
| rotor speed standard deviation | -47.4% | -13.8% | -44.0% | -14.1% | -44.6% |
| tower-base fore-aft DEL | -4.3% | -1.0% | -4.1% | -1.1% | -4.3% |
| blade-root flap-wise DEL | -3.1% | 0.4% | -2.7% | 0.2% | -2.9% |

**Table 1.** Control benefits of feedforward-feedback combined controllers relative to using feedback-only controllers for: a single-beam lidar in the spinner and on the nacelle both using a CW and a pulsed system, and a 4-beam CW lidar on the nacelle at a mean wind speed of $18$ m s$^{-1}$.

## 6 Discussions

The goal of this study is to demonstrate that a single-beam lidar mounted in the spinner increases the performance of feedforward control compared to the same lidar mounted on the nacelle. The study optimizes the lidar scanning configurations for the

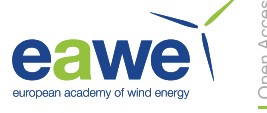
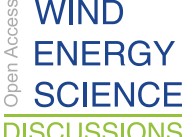

**Figure 7.** Spectra of the rotor speed (RotSpeed), tower-base fore-aft (TwBsMyt) and blade-root flap-wise bending moments (RootMyc1) with feedback-only (FB) and the feedback-feedforward combined (FBFF) controller using the optimal single-beam CW lidar (a)(c)(e) in the spinner and (b)(d)(f) on the nacelle at a mean wind speed of $18 \text{ ms}^{-1}$. Simulation results are the average using 21 wind field realizations. Some relevant structural frequencies are marked.





best wind preview quality considering the longitudinal wind evolution in the wind field. The optimum configurations for both

CW and pulsed lidars are selected for a mean wind speed of $18\ \mathrm{ms}^{-1}$.

The strength of wind evolution is one of the factors that affect the optimal lidar scanning configuration. Other factors include the number and the location of measurements, the turbulence spectra, and the severity of contamination by the transverse velocity components, which is affected by the lidar beam directions (Guo et al., 2022a). The smaller the beam opening angle, the smaller the contribution of the transverse velocity components to the radial velocity. To reveal the impact of turbulence

evolution, Figure 8 shows the optimization results of the CW and pulsed lidars when the evolution is neglected and Taylor's frozen turbulence hypothesis is applied. Compared to those shown in Figure 5, the maximum achievable MCBs of both lidars are overestimated. For the CW lidar, assuming frozen turbulence does not change the shape of the MCB curve. This is expected because the probe volume of a CW lidar increases quadratically with the focus distance, which plays a more important role in determining the MCBs than the turbulence evolution. As for the pulsed lidar whose probe volume does not change with

the measurement range, the highest MCB is reached at a further measurement distance at $x = 270$ m and a smaller opening angle $\phi = 15°$ compared to the optimum in Figure 5. The resulting lidar scan radius remains at $\approx 0.6R$. Owing to the rotor shaft tilt angle, measuring too far away from the rotor causes the lidar scanning area to be easily out of the rotor swept area. Therefore, the MCB decreases from the optimum point when the lidar measures at $x = 270$ m with a wider opening angle or with $\phi = 15°$ at a further measurement distance. In summary, the neglection of wind evolution can result in an overestimation

of MCB, wrong selection of the optimum lidar configuration, and eventually the underperformance of the feedforward control, especially in unstable atmospheric conditions.

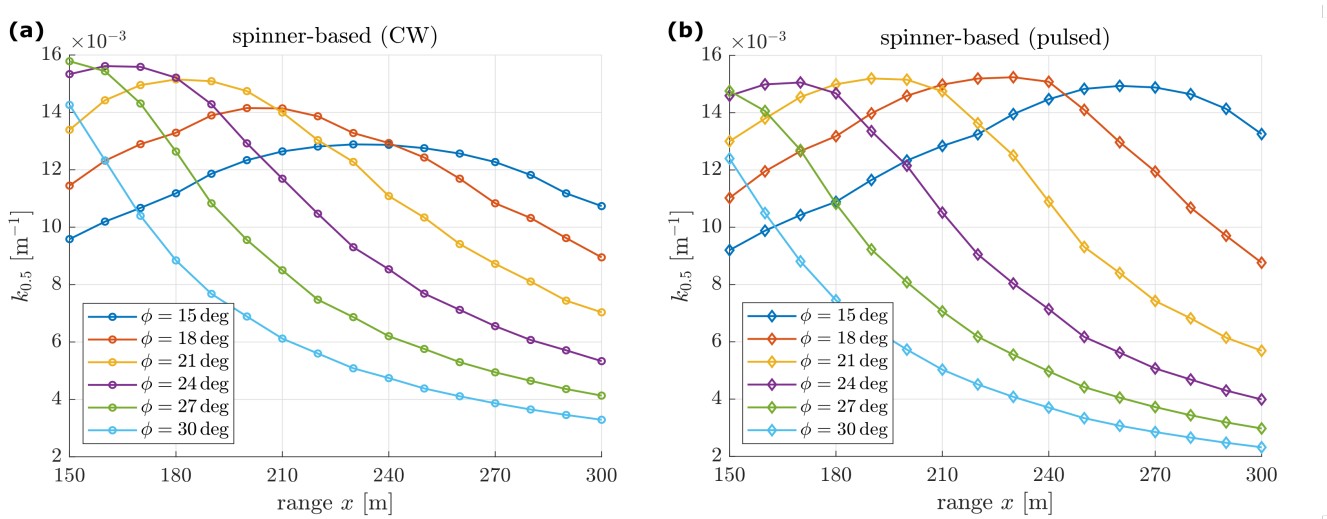

**Figure 8.** Optimization of the range $x$ and half-cone opening angle $\phi$ of the spinner-based lidar when wind evolution is neglected.

As mentioned in Section 4, for higher wind speeds, larger measurement ranges are needed for both CW and pulsed lidars so that the controllers have enough time to react to the wind disturbance. Further work needs to be done with the full wind speed





range to decide the best scanning configuration of the single-beam lidar in the spinner. Also, the controller performances can
be influenced by turbulence conditions. Only neutral atmospheric stability is considered in this work. Guo et al. (2023) showed
that the control benefit is at its highest in unstable, middle in neutral, and lowest in stable atmospheric conditions.

## 7 Conclusion and Outlook

A single-beam Doppler lidar is flexible and low in cost. Using the single-beam lidar in the spinner, the lidar can rotate with the
rotor at an almost steady rotational speed when the turbine operates at above-rated wind speeds and scans a good portion of the
inflow to the rotor disk. Also, the spinner-based lidar can have a view of the inflow without periodic blockage by the running
blades, which improves the lidar data availability.

Based on a coherence model of the lidar-rotor REWS using 4D Mann turbulence model, this work optimizes the scanning
configurations (i.e., measurement range and the half-cone opening angle) of the spinner-mounted single-beam lidar in a CW
and a pulsed system, respectively, at a single wind speed of $18\ \mathrm{ms^{-1}}$ for the bottom-fixed IEA 15-MW wind turbine. The
optimum configurations of the two types of lidars are different due to the spatial averaging effect of their probe volumes, but
they both result in a scan radius of approximately $0.6$ of the turbine radius. The optimum configurations of both types of lidars
give a MCB of around $0.014\ \mathrm{m^{-1}}$, which corresponds to the smallest detectable eddy size of $1.87D$. Large lidar measurement
ranges are needed to ensure the turbine controllers have enough time to react to the wind disturbance over the full wind speed
range, which slightly reduces the MCB.

Using time-domain simulations and 4D Mann turbulence wind fields in the neutral condition, the benefits of regulating rotor
speed variation and reducing fatigue loads on the tower and blades using the feedforward controller and the spinner-based
single-beam lidar are evaluated for the reference turbine at a single wind speed of $18\ \mathrm{ms^{-1}}$. Results are compared against a
single-beam and a 4-beam nacelle-based lidar. The control benefits using the optimized spinner-based configurations of both
CW and pulsed lidars are much higher than the single-beam nacelle lidar, and they are on a similar level to the 4-beam nacelle
lidar.

For future work, full wind speed ranges up to the wind turbine cut-off wind speed should be considered to select the optimum
scanning trajectory of the spinner-based single-beam lidar for the IEA 15-MW wind turbine. The pulsed lidar could potentially
deliver a better wind preview signal than the one shown in this work when measurements at multiple measurement ranges
are combined. In addition, more reductions in fatigue loads could be achieved by optimizing the parameters of the feedback
controller. In the future, more than one single-beam lidar can be used in the spinner to add redundancy to the system, meanwhile,
having the possibility to achieve a shorter full scan time or multi-plane measurements simultaneously even with CW lidar
systems.

*Code availability.* The 4D Mann turbulence generator is accessible via https://github.com/MSCA-LIKE/4D-Mann-Turbulence-Generator.
The source code of the ROSCO controller can be found by https://github.com/NREL/ROSCO, version 2.6.0. The source code and compiled



DLLs for the ROSCO feedback controller, the lidar data processing and a collective pitch feedforward controller, and a wrapper DLL are accessible via https://github.com/MSCA-LIKE/Baseline-Lidar-assisted-Controller.

*Author contributions.* All authors participated in the conceptualization and design of the work. FG and WF derived the REWS coherence model. DS and WF designed the lidar-data-processing unit for the spinner-based lidar and did load characterization. WF did lidar optimizations, performed time-domain simulations and wrote the manuscript. All authors supported the whole analysis and reviewed and edited the
manuscript.

*Competing interests.* The authors have no competing interests to declare.

*Acknowledgements.* This study is funded by the European Union's Horizon 2020 research and innovation program under the Marie Sklodowska-Curie grant agreement No. 858358 (LIKE – Lidar Knowledge Europe, H2020-MSCA-ITN-2019). The authors would like to thank Prof. Jakob
Mann for the discussion on the modeling of wind evolution and REWS spectra.





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
