# Peer review of "Feedforward pitch control for a 15-MW wind turbine using a spinner-mounted single-beam lidar"

_Wind Energy Science, 2023_

## Author Response (AR1)

**Response to Referee 1**

Dear Alan Wai Hou Lio,

Thank you for your general comments on our work, which we consider very important to improve the manuscript. Here is our response to each of your comments. Comments from the reviewer are reported in black and followed by our answers in blue.

Besides the changes made based on your comments, we have made two additional changes:

1. We removed the data tips in Figure 5 (b) and (d) for the clarity of the figure.
2. We deleted "transfer function" in L361 of the revised manuscript. Now the sentence is "Figure 6 compares the REWS coherence from time-domain simulations and those calculated in the frequency domain using the method presented in Section 2.4. " because no transfer function was shown in Figure 6.

Best regards,
The authors
* * *
**Main comments:**

1. "In this work, a single-beam continuous-wave or a pulsed lidar system is simulated in the spinner of a bottom-fixed IEA 15 MW wind turbine." - Maybe rephrase this because, from my understanding, both continuous-wave and pulsed lidar were considered in this work.

   *Thank you for the advice. We changed the sentence to "In this work, a single-beam lidar is simulated in the spinner of a bottom-fixed IEA 15 MW wind turbine. Both continuous-wave and pulsed lidar systems are considered." (L3)*

2. Page 11. "(in our case 0.025 Hz)" How did you know this number?

   *For the design of a low-pass filter in the lidar data processing unit, we selected the $f\_delay$ around the frequency, in which the rotor speed simulated with only the feedback controller fluctuates the most (i.e., the rotor speed spectrum has its highest energy). We took $f\_delay = 0.025$ Hz from a previous work [2] and didn't change it. We added the reference on Page 11 (L162). The $f\_delay$ could have an impact on the $T\_filter$ (see Eq. (30)), but the impact is only marginal for the spinner-based lidars (with a $f\_cutoff$ around 0.1597 $rad\ s^{-1}$ in this work).*

3. Page 11. Equation 31. Please state the assumptions. For example, it is a direct-drive design (no gearbox ratio), with no electrical conversion efficiency.

   *Thank you for pointing out that. In the revised manuscript (L270), we noted that Eq. (31) is "a reduced-order model of the direct-drive IEA 15-MW wind turbine [1] with a single rotor rotation DOF".*

4.  Page 11 Line 279. "Thus, a feedforward pitch rate θ˙FF can be calculated using the derivation of the static pitch curve (see Schlipf, 2016… )" I think in David's work, he computed the pitch rate using the derivative (d \theta/dv) and the derivative of the wind speed, and also imposed a limit on the d\theta/dv. Did you do the same in this article?

    *In this article, we only calculated the feedforward pitch angle by interpolation of the static pitch curve and then made a simple time derivative to compute the feedforward pitch rate. We elaborated in L282-286 of the revised manuscript: "Here, we only use the static pitch curve above rated wind. The feedforward pitch angle is obtained by interpolating the static pitch curve in every simulation time step. As described in Schlipf (2016) (see Chapter 6.1.1 for more details), using a feedforward pitch rate $\dot{\theta}_{FF}$ instead of the feedforward pitch angle has advantages for the implementation of the feedback-feedforward combined controller. Therefore, we use a simple time derivative of the feedforward pitch angle to obtain the feedforward pitch rate and then add the pitch rate to the integrator input of the feedback controller.*

5.  Page 12. Line 297. "… for frequencies lower than 0.04 Hz …" How did you end up in this number? Can you elaborate?
    *In the revised manuscript, we changed the last paragraph (L303-305) of Section 3.4, "The pitch actuator is modelled as a second-order damper system with the cut-off frequency of $1.5708 \ rad \ s^{-1}$ and the damping ratio of 0.707, which are based on the values of the ROSCO designed for IEA 15-MW wind turbine [4]". Therefore, The phase response of the dynamic system can be computed, from which the pitch actuation time can be obtained. Figure 1 below shows the magnitude, phase and time delay of the pitch actuator system. We found that $T_{pitch}$ is around 0.9 for frequencies lower than 0.04 Hz.*

[Figure]

*Figure 1: Magnitude, phase and time delay of the IEA 15-MW turbine pitch actuator over frequency.*

6. Page 12. Equation 33 and 34. The integral bound is "t" and the variable of integration is also "t" as in the symbol "dt", which is not right. Maybe change the differential of the variable to "d\tau".

   *Thanks for pointing that out. You are right. We changed the symbol "dt" to "d\tau" in Eq. (33) and Eq. (34).*

7. Page 12, Line 290. "…kp is the proportional gain, KI the integral gain…" Why is it that one is capitalised while the other is not?

   *Thanks for pointing that out. We changed "$K_I$" to "$k_i$" in the revised manuscript (Eq.(33) and L296).*

8. Page 12 Line 290. "ΔΩ = ΩG,rated−ΩG" It should be \Delta Omega = Omega_G - Omgea_G,rated, unless the Kp and Ki are negative.

   *You are right. We corrected the expression in L296 of the revised manuscript.*

9. Section 4. This optimisation of LiDAR configuration was only conducted for one turbulence intensity? and one mean wind speed with 21 seeds?

   *The lidar configurations were optimized in the frequency domain using the analytical coherence model $\gamma_{RL}^2(k_1)$ of the REWS (see Eq.(26)) without running aero-elastic simulations. We assumed one set of turbulence parameters for a neutral turbulence condition and a mean wind speed of 18 $ms^{-1}$.*

10. Page 15. Line 366. "GΩuLL is the closed-loop transfer function from the REWS to the rotor speed," Is the transfer function obtained from OpenFAST or just a simplified 1DOF drive-train/rotor model with the PI controller? If it's from OpenFAST, does it include all the structural dynamics and how many system states are there?

    *In the revised manuscript L380, we changed the text to "$G_{\Omega u_{LL}}$ is the closed-loop transfer function from the REWS to the rotor speed, which is obtained from the simplified 1DOF rotor model with the feedback (PI) controller, low pass filter and the pitch actuator [5]."*

11. Section 5. Regarding the simulation, was the LiDAR implementation considered the blockage of rotating blades in OpenFAST, which was stated as the benefit of the proposed LiDAR in the abstract? I wonder how much having a clear view (without the blockage) would influence the MCB.

    *Our simulations did not consider the blockage of the rotating blades. As shown in Section 5.4 (Figure 9) in [3], the blade blockage does not have a significant impact and thus we neglected it for simplicity.*

**References:**

[1] E. Gaertner *et al.*, "Definition of the IEA Wind 15-Megawatt Offshore Reference Wind Turbine Technical Report," 2020, Accessed: Dec. 20, 2022. [Online]. Available: www.nrel.gov/publications.

[2] D. Schlipf, F. Guo, S. Raach and F. Lemmer, "A Tutorial on Lidar-Assisted Control for Floating Offshore Wind Turbines," 2023 American Control Conference (ACC), San Diego, CA, USA, 2023, pp. 2536-2541, doi: 10.23919/ACC55779.2023.10156419.

[3] F. Guo, D. Schlipf, H. Zhu, A. Platt, P. W. Cheng, and F. Thomas, "Updates on the OpenFAST Lidar Simulator," J. Phys. Conf. Ser., vol. 2265, no. 4, p. 042030, May 2022, doi: 10.1088/1742-6596/2265/4/042030.

[4] N. J. Abbas, D. S. Zalkind, L. Pao, and A. Wright, "A reference open-source controller for fixed and floating offshore wind turbines," *Wind Energy Sci.*, vol. 7, no. 1, pp. 53–73, Jan. 2022, doi: 10.5194/WES-7-53-2022.

[5] D. Schlipf, E. Simley, F. Lemmer (né Sandner), L. Pao, and P. W. Cheng, "Collective pitch feedforward control of floating wind turbines using lidar," *J. Ocean Wind Energy*, vol. 2, no. 4, pp. 223–230, 2015, doi: 10.17736/jowe.2015.arr04.

**Response to Referee 2**

Dear referee,

Thank you for your general comments on our work, which we consider very important to improve the manuscript. Here is our response to each of your comments. Comments from the reviewer are reported in black and followed by our answers in blue.

Besides the changes made based on your comments, we have made two additional changes:

3. We removed the data tips in Figure 5 (b) and (d) for the clarity of the figure.
4. We deleted "transfer function" in L361 of the revised manuscript. Now the sentence is "Figure 6 compares the REWS coherence from time-domain simulations and those calculated in the frequency domain using the method presented in Section 2.4. " because no transfer function was shown in Figure 6.

Best regards,
The authors
* * *
**Main comments:**

1. Line 125: It is not clear for people out of the domain what are the x, y and z axis, you might refer to figure 1 at least and tell where does the wind come from on it

   *Thanks for the suggestion. In Section 2.3 (L122-124) of the revised manuscript, we added "Figure 1 shows the scanning trajectory of the single-beam lidar in the spinner of the 15-MW turbine, where x-, y- and z-axis describe the coordinates of the three-dimensional wind field, as introduced in Section 2.1. The mean wind direction is along the x-axis."*

2. Line 135: The equation is wrong, it should be 60/7.56= 8s

   *Thanks for pointing that out. You are right. We corrected the sentence (L136): "Therefore, the single-beam lidar needs around 8 s ($2\pi/7.56$ rpm) to complete a full scan."*

3. Line 159: I guess that you mean micrometer here, could you please make the \mu stick to the m?

   *You are right. We have adjusted it.*

4. Line 260: Can you explain further where this 0.025Hz is coming from?

   *For the design of a low-pass filter in the lidar data processing unit, we selected the $f\_delay$ around the frequency, in which the rotor speed simulated with only the feedback controller fluctuates the most (i.e., the rotor speed spectrum has its highest energy). We took $f\_delay = 0.025$ Hz from a previous work [1] and didn't change it. We added the reference on Page 11 (L162). The $f\_delay$ could have an impact on the $T\_filter$ (see Eq. (30)), but the impact is only marginal for the spinner-based lidars (with a $f\_cutoff$ around 0.1597 $rad\ s^{-1}$ in this work).*

5.  Line 309: Can you write down explicitly the optimization that is considered here and refer to this equation

    *Thanks for the suggestion. We described the optimization problem with a cost function and added in Section 4 (L317-320) of the revised manuscript: "The optimization problem can be formulated as*

    $$\underset{x,\phi}{maximize} \ \ MCB$$

    $$\text{Subject to: } T_{buffer} \geq 0,$$

    *which uses the measurement range x and the lidar half-cone opening angle $\phi$ as the optimization variables, MCB as the cost function, and a positive buffer time as the constraint. The optimization problem is solved by brutal-force optimization."*

6.  Line 322-323: The sentence "This is expected since the probe volume filtering effect becomes more influential for CW lidars the further the measurement range" is not clear.

    *L333-L336 of the revised manuscript, we elaborated on this "This is expected since the further the CW lidar measures, the larger the lidar probe volume, whereas the probe volume of the pulse lidar does not change with its measurement range. The probe volume filtering effect (along with other effects, such as wind evolution) contributes to the decrease of MCB with increasing measurement distance."*

7.  Line 377: Rainflow is in a single word.

    *Thanks for pointing that out. We have corrected the word in L391.*

**References:**

[1] D. Schlipf, F. Guo, S. Raach and F. Lemmer, "A Tutorial on Lidar-Assisted Control for Floating Offshore Wind Turbines," 2023 American Control Conference (ACC), San Diego, CA, USA, 2023, pp. 2536-2541, doi: 10.23919/ACC55779.2023.10156419.